# Sparsified federated learning with differential privacy for intrusion detection in VANETs based on Fisher Information Matrix

**Rui Chen** *, Xiaoyu Chen, Jing Zhao

School of Software Technology, Dalian University of Technology, Dalian, Liaoning, China

* chenrui_dut@163.com

**Data Availability Statement:** The datasets supporting the conclusions of this study are available in the following publicly accessible repositories: CIC-IDS-2017, which can be found at https://doi.org/10.5220/0006639801080116

## Abstract

With the continuous development of vehicular ad hoc networks (VANET) security, using federated learning (FL) to deploy intrusion detection models in VANET has attracted considerable attention. Compared to conventional centralized learning, FL retains local training private data, thus protecting privacy. However, sensitive information about the training data can still be inferred from the shared model parameters in FL. Differential privacy (DP) is sophisticated technique to mitigate such attacks. A key challenge of implementing DP in FL is that non-selectively adding DP noise can adversely affect model accuracy, while having many perturbed parameters also increases privacy budget consumption and communication costs for detection models. To address this challenge, we propose FFIDS, a FL algorithm integrating model parameter pruning with differential privacy. It employs a parameter pruning technique based on the Fisher Information Matrix to reduce the privacy budget consumption per iteration while ensuring no accuracy loss. Specifically, FFIDS evaluates parameter importance and prunes unimportant parameters to generate compact sub-models, while recording the positions of parameters in each sub-model. This not only reduces model size to lower communication costs, but also maintains accuracy stability. DP noise is then added to the sub-models. By not perturbing unimportant parameters, more budget can be reserved to retain important parameters for more iterations. Finally, the server can promptly recover the sub-models using the parameter position information and complete aggregation. Extensive experiments on two public datasets and two F2MD simulation datasets have validated the utility and superior performance of the FFIDS algorithm.

## 1 Introduction

With the rapid development of vehicular ad hoc networks (VANET), the proliferation of diverse in-vehicle communication devices has led to massive data generation through their rich sensing, computing, and storage capabilities [1, 2]. Currently, such huge amounts of data face varying degrees and types of security threats that could jeopardize vehicles on the road and human lives. Fortunately, intrusion detection techniques can isolate compromised networks or switch vehicles into a safe mode during driving, thus reducing safety threats. Since

(referenced as [17]), CAN-Intrusion-Datasets, available at https://doi.org/10.1109/PST.2017.00017 (referenced as [16]). Additionally, the VeReMi and BSMList datasets are available at https://zenodo.org/doi/10.5281/zenodo.10785299.

**Funding:** The author(s) received no specific funding for this work.

**Competing interests:** The authors have declared that no competing interests exist.

deep learning models can process massive data and learn from heterogeneous sources with high accuracy, they hold promise for enabling VANET devices to detect various threats in VANET [3, 4]. However, conventional centralized machine learning paradigms require transmitting all raw data to the cloud before model training, incurring prohibitive communication costs and serious privacy risks [5].

In the field of VANET security, the application of federated learning (FL) has emerged as a promising approach, garnering increasing attention for its proficiency in enhancing communication efficiency and fortifying privacy. FL employs a central server to orchestrate a collaborative effort among clients to train a detection model across several global iterations [5]. In each iteration, the server distributes the current model parameters to a subset of participating clients who then execute local training on their confidential datasets. Subsequently, these clients transmit their locally updated models to the server, where they are amalgamated to refine the global model. Despite the strengths of FL, it does not provide an absolute privacy shield, as the potential for sensitive training data to be deduced from the shared model parameters remains—a vulnerability exposed by various inference attacks [6]. Notably, member inference attacks can exploit an attack model to ascertain whether a specific sample contributed to the training of the target model, based on knowledge of the input sample and the model itself [7, 8]. Furthermore, through model inversion attacks, adversaries are capable of reconstructing original data representations for certain classes by training an inversion model that exploits the correlations discernible between the inputs and outputs of the target model, provided they have access to the model and corresponding class labels [9].

To further enhance privacy protection, differential privacy (DP) has become the de facto standard for data privacy, and incorporating DP into VANET intrusion detection can provide rigorous privacy guarantees against adversaries with arbitrary auxiliary information [10, 11]. For privacy preservation in FL, clients add DP noise to local model parameters before sharing them with the central server. Meanwhile, malicious attackers cannot easily recover the noisy local models. Unfortunately, implementing DP in the shared model parameters of FL while maintaining high model accuracy faces several major challenges: First, FL is an iterative learning process where model updates are exchanged over multiple rounds. Most existing studies only incorporate a noise addition module to model parameters without selective perturbation, adding noise indiscriminately to all uploaded parameters [5, 12]. This excessively consumes the privacy budget. Since the budget signifies the maximum tolerable privacy loss for clients, excessive consumption significantly harms model accuracy. Second, the non-selectively added DP noise scales linearly with model size. For VANET intrusion detection models which can be very large, this undoubtedly incurs immense communication costs in FL [12]. Third, the challenge of applying federated learning with differential privacy (FLDP) to VANET intrusion detection is that model training is highly complex, yet the privacy budget is extremely limited. This can result in very poor detection performance. Prior work shows most client updates are unimportant parameters with values close to zero. Adding noise to such unimportant parameters before uploading for aggregation wastes the privacy budget. Intuitively, clients should only protect the most important parameters within range (far from zero) to reduce budget consumption [10, 13]. However, directly selecting certain parameters to protect poses a tremendous challenge for server aggregation, as it has to further infer parameter positions while still returning full parameters in the end, meaning only the uploading stage is reduced. Therefore, we need to design algorithms that concurrently preserve important parameters and the selection scheme.

To address the existing challenges, we propose a novel FL algorithm called FFIDS that integrates model parameter pruning with DP. By pruning unimportant parameters, FFIDS effectively reduces the privacy budget consumption during iterative training while ensuring model

accuracy and saving communication costs. Specifically, FFIDS leverages the Fisher Information Matrix [14, 15] to evaluate the importance of client model parameters. Parameters with near-zero values that contribute minimally to parameter aggregation on the central server may undergo significant changes under DP noise. Hence, we identify and prune these near-zero weighted parameters, which not only decreases the privacy budget consumption per iteration, but also guarantees no loss in model accuracy. Next, we add DP noise to the pruned sub-models, fulfilling the privacy requirements of FL and making it extremely difficult for adversaries to infer client raw data from the parameters. When clients upload the perturbed sub-models, they also share a position matrix. This binary matrix flags whether each parameter is pruned or not. Since each element occupies only 1 bit and the overall matrix is around 3% of the original model size, while pruning ratio far exceeds 3%, both model uploading and distribution by the server involve only the pruned models. This simultaneously expedites server aggregation and substantially reduces communication costs in FLDP. We have conducted extensive experiments on public datasets [16, 17] and F2MD [18] simulated datasets to validate the superiority of FFIDS in maintaining model accuracy over existing baselines.

Above all, we are the major contribution of this work is as follows:

- We pioneer the introduction of the Fisher Information Matrix into the field of FLDP and innovatively propose the FFIDS algorithm. It aims to significantly improve model training with lower communication costs while reducing privacy budget consumption and maintaining detection accuracy.

- By applying FFIDS to federated learning with differential privacy protection, we theoretically analyze its remarkable efficacy in saving privacy budgets, thereby more effectively balancing model privacy and utility.

- To effectively harness the Fisher Information Matrix under the FL framework, we design a parameter pruning and recovery scheme based on importance evaluation of model parameters. This constructs a parameter position matrix by dropping insignificant parameters according to the importance metric, thus assisting the central server in accurately recovering the pruned parameter positions.

- Experiments on public datasets and F2MD simulated datasets validate the superior performance of FFIDS in terms of model accuracy and loss function optimization. Meanwhile, the algorithm also excels in reducing communication and computation costs as well as privacy budget consumption, possessing significant competitive advantages.

## 2 Background and literature review

### 2.1 Intrusion detection in VANET

Cybersecurity challenges are increasingly jeopardizing the integrity of intelligent vehicular ad hoc networks (VANET), with mounting evidence from security researchers highlighting vulnerabilities in actual vehicle systems [19, 20]. The risk landscape is vast, with millions of vehicles susceptible to an array of safety [21] threats [22]. A notable instance was when Miller et al. [22] exploited open Wi-Fi ports to compromise the In-Vehicle Network (IVN) of a Jeep Cherokee, subsequently rewriting the ECU firmware. To bolster the resilience of smart car networks against such cyber-attacks, ongoing research endeavors are focused on pioneering new methodologies and algorithms.

Contemporary studies have put forth an assortment of intrusion detection strategies, including fingerprint recognition and parameter monitoring, tailored for automotive

Controller Area Network (CAN) systems. Despite these advances, each technique comes with inherent constraints, making the realization of a fully secure IVN system elusive [23]. Garg et al. [24] introduced a probabilistic data structure-based vehicle network IDS, which when benchmarked against ANN and SVM classifiers, showcased superior authentication accuracy and adaptability within high-velocity Internet of Vehicles (IoV) networks [25]. Nevertheless, these solutions are predicated on traditional machine learning frameworks. With the advent of sophisticated deep learning methodologies, their application in IoV intrusion detection is gaining traction.

Almutlaq et al. [26] delved into the use of interpretable neural networks for intrusion detection, validating their approach across diverse datasets and achieving commendable accuracy and performance metrics. Xie [27] leveraged the potential of Generative Adversarial Networks (GANs) for crafting viable attack simulations to enrich training datasets in intrusion detection contexts. Ashraf et al. [28], as well as Lin et al., employed encoders to condense dataset features, followed by time series models for detecting intrusions. Li et al. [29] devised a multi-tiered hybrid intrusion detection system adept at recognizing both known and novel threats. This system utilizes transfer learning to encode CAN *DATA* into a feature matrix and employs a Convolutional Neural Network (CNN) for attack detection. Despite their efficacy in identifying various cyber threats, including malware and malicious activities, deep learning models face a common hurdle: the scarcity of attack samples during training, which pales in comparison to the abundance of normal samples. This imbalance hampers the models' ability to perform precise intrusion detection. Furthermore, contemporary automotive electronic systems often require extensive time for modifications and upgrades to fortify security measures.

## 2.2 Enhanced intrusion detection in VANET with privacy-preserving federated learning

FL [30] has revolutionized the field of machine learning by addressing the challenges of implementing deep learning in privacy-sensitive environments. In FL architectures, a central server facilitates the collaborative training of intrusion detection models by connecting to multiple vehicles through advanced communication protocols such as 5G or IEEE 802.11p. This decentralization is a marked departure from traditional deep learning paradigms that rely on centralized data collection. The distributed nature of FL ensures that the data remains on local devices, with a global server orchestrating the process by distributing the initial training models and defining aggregation algorithms [7–9]. This minimizes direct access to training data, thereby maintaining privacy and reducing the infrastructural costs associated with centralized models. Nevertheless, this setup is not without its challenges; bandwidth constraints and privacy concerns arise during the transmission of gradient updates. In our study, we have enhanced the FL framework with a Laplacian noise mechanism [11] to safeguard against reverse engineering of model parameters. To further ensure privacy, our communication protocol strictly transmits only model-related parameters, prohibiting the exchange of raw data and sensitive statistics [31]. FedAvg [30] remains the cornerstone aggregation algorithm in FL, predicated on the assumption of uniform data distribution across local devices.

Privacy preservation in FL is critical, as highlighted by Zhao et al. [10], who demonstrated that an adversary could potentially reconstruct client data from leaked gradients. Dwork's introduction of differential privacy (DP) [32] has been pivotal in protecting data within machine learning. Abadi et al. [33] further refined this concept with the differential privacy stochastic gradient descent (DP-SGD) algorithm, providing a framework for quantifying privacy loss. Local Differential Privacy (LDP) [34, 35] has been employed to anonymously

aggregate user data, and Chen et al. [36] have proposed personalized LDP to enhance data utility, allowing users to tailor their privacy settings.

Despite the benefits of DP in securing data, it can compromise model accuracy due to the injected noise. Comprehensive studies in the domain of the VANET have underscored the detrimental impact of DP noise on the efficacy of machine learning models [37]. To address the limitations of DP in FL, recent works [38] suggest returning only the most significant gradients in each query, thereby minimizing the expenditure of the privacy budget. Yet, this approach often leads to suboptimal training efficiency due to FL's inherent communication delays and the high dimensionality of learning models [39]. Innovative solutions such as Sparse Vector Technique (SVT) have been proposed [12] where gradients are only uploaded when there is a significant change in the loss function.

Building on these foundations, our work pioneers the use of the Fisher Information Matrix to gauge the importance of each gradient, aiming to reduce the consumption of privacy budgets for gradients nearing zero. This novel approach significantly mitigates the influence of noise in FLDP, while also lowering the risk of data leakage from model inversion attacks. Our model demonstrates superior adaptation to real-world intrusion detection scenarios and, when compared to other machine learning and deep learning-based intrusion detection systems, our IDS exhibits enhanced accuracy, making it a viable solution for VANET intrusion detection systems.

## 3 Preliminaries

In this section, we introduce preliminaries on DP and the Fisher Information Matrix.

### 3.1 Differential privacy

To ensure the protection of data privacy, Differential Privacy (DP) [40] technology plays a critical role in the query process of dataset $D$ by adding noise perturbation to the query results. The formal definition of $(\epsilon, \delta)$-DP is as follows:

**Definition 1:** Let $D$ and $D'$ be two adjacent datasets that satisfy $\|D - D'\|_1 \leq 1$. If there exists a randomized algorithm $R$ such that for all adjacent datasets $D$ and $D'$, and for all possible output subsets $S \subseteq Range(R)$ of algorithm $R$, the following inequality holds:

$$Pr\{R(D) \in S\} \leq exp(\epsilon) \times Pr\{R(D') \in S\} + \delta \tag{1}$$

Where $Range(R)$ denotes the set of all possible outcomes of the algorithm $R$. In the definition above, the parameter pair $(\epsilon, \delta)$ is referred to as the privacy budget. The smaller the values of the privacy budget, the larger the variance of the noise added, and the higher the level of privacy protection provided by algorithm $R$. When $\delta = 0$, the algorithm $R$ satisfies $\epsilon$-DP.

Let $w$ be the input parameter for the query, and $s(w, D)$ be the precise query result on dataset $D$ with respect to input $w$. Define sensitivity $\Delta s$ as the maximization of the $L_1$ norm difference of the query function $s$ across all inputs $w$ and datasets $D$, i.e., $\Delta s = \max_{\forall w, D} \|s(w, D) - s(w, D')\|_1$. Where $D$ and $D'$ are two adjacent datasets that satisfy $\|D - D'\|_1 \leq 1$. The Laplace mechanism achieves $\epsilon$-DP protection by perturbing the query result with noise conforming to the Laplace distribution.

**Theorem 1:** Given a dataset $D$ and a query parameter $w$, a Laplace mechanism satisfying $\epsilon$-DP will perturb the query result $a_L$ by adding noise as follows:

$$a_L(w, \epsilon) = s(w, D) + Z \tag{2}$$

where $Z$ represents the noise generated by the Laplace distribution, with the probability density function $Pr(Z) = \frac{\epsilon}{2\Delta s} \exp\left(-\frac{\epsilon|Z|}{\Delta s}\right)$, which can also be denoted as $Z \sim \text{Lap}\left(\frac{\Delta s}{\epsilon}\right)$. According to

Theorem 1, the consumption of the privacy budget is closely associated with the behavior of answering the query.

## 3.2 Gradient pruning for Fisher Information Matrix

To avoid expending an excessive privacy budget on protecting negligible gradients, it is necessary to prune those gradients deemed insignificant, that is, to refrain from uploading them to the central server for aggregation. Intuitively, if a gradient is very close to zero, its contribution to the model aggregation process can be considered negligible. It is only meaningful to aggregate those gradients that are significantly greater than zero.

Theorem 1 addresses the consumption of privacy budget in the context of a single query. When differential privacy's composability principle is applied, the privacy budget's depletion is cumulative for multiple queries, increasing linearly with the number of queries. To establish a general framework, suppose the FL model is characterized by a dimensionality $d$. A single update from a vehicle-side client, involving $d$ parameters, is analogous to answering $d$ individual queries. Let us define the vector $w$ as the set of input parameters across $d$ queries, where $s(w, D) \in \mathbb{R}^d$ represents the exact query outputs for input $w$ against the dataset $D$. The notation $[i]$ refers to the $i$-th element of any given vector. Applying the Laplace mechanism to answer $d$ queries, which is defined by the process $R_L(w, \epsilon) = s(w, D) + Z$, expends a total privacy budget of $d\epsilon$. In this expression, $Z$ denotes a $d$-dimensional vector of noise, with each component $Z[i]$ (for all $1 \leq i \leq d$) being independently and identically distributed (IID) according to the Laplace distribution $\text{Lap}\left(\frac{\Delta s}{\epsilon}\right)$.

To curtail the privacy budget expenditure in the context of multiple queries, while proposals exist that leverage a pruning mechanism informed by the Fisher Information Matrix, these have not been finely tuned for the specific nuances of FL. In a more detailed view, consider a training dataset comprising pairs $(x_1, y_1), (x_2, y_2), \ldots, (x_n, y_n)$ and a model $f(y|x;w)$ with a loss function denoted by $Loss(f(y|x;w))$. Here, $w$ symbolizes the entirety of the model's parameters, with $w[i]$ representing the $i$-th parameter. The Fisher information is utilized to gauge the significance of each parameter, and is formally defined as follows:

$$F_i = \sum_{j=1}^{n} \left( \frac{\partial Loss(f(y_j|x_j; w))}{\partial w[i]} \right) \tag{3}$$

Upon assessing the significance of model parameters, clients are tasked with independently determining a suitable pruning strategy that aligns with their local data distribution, communication cost, and computational capabilities. Subsequently, they prioritize and transmit the most crucial parameters to the server for amalgamation. During the model update upload, clients selectively refine only the parameters exempted by the pruning algorithm. This method streamlines the optimization process by focusing on fewer parameters, consequently mitigating the overfitting risk. Moreover, it substantially diminishes the communication cost in the federated learning workflow, leading to improved efficiency and resource management.

## 4 FFIDS algorithm design

In this section, we integrate the Fisher Information Matrix into the FLDP framework to bolster data privacy safeguards while maintaining the accuracy of the FFIDS algorithm. Furthermore, we conduct a thorough analysis of the privacy protection afforded by our proposed algorithm, juxtaposing it with state-of-the-art technologies to highlight its superior advantages.

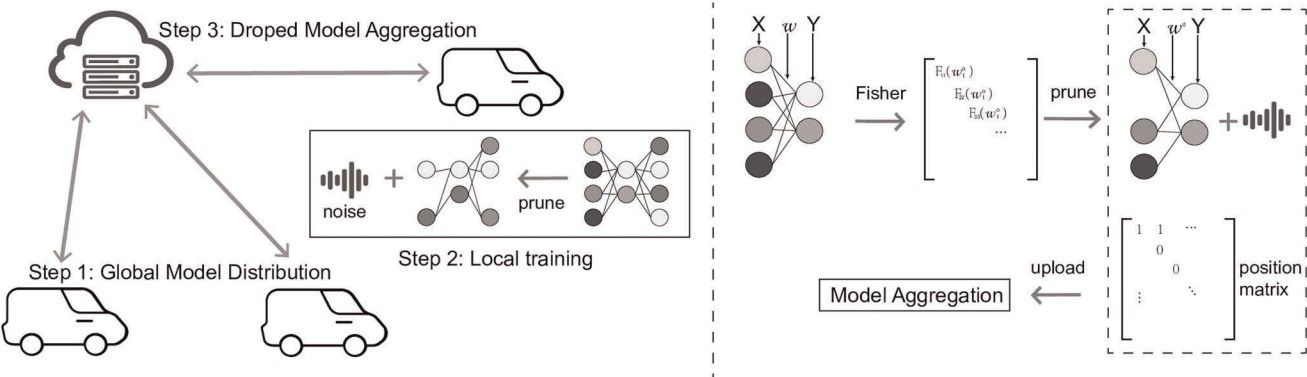

**Fig 1. FLDP intrusion detection framework and model pruning process.**

## 4.1 FLDP framework

To enhance comprehension of the FLDP framework within VANET, we refer to left image of Fig 1, which dissects the algorithmic framework into two principal components: the server-side and the vehicle-side (client).

Server: The server embodies a computational hub equipped with substantial processing capabilities, typically perceived as a semi-honest entity. It faithfully executes training tasks yet harbors an inherent curiosity about the clients' private data. Within the conventional federated learning paradigm, the server singularly orchestrates the initiation of training, model aggregation, dissemination, and other pivotal functions.

Client: On the vehicle-side, the client is replete with sensitive data but is constrained by limited computational capacity, rendering it less suited for conducting extensive model training tasks.

Subsequently, we can present a general procedure for federated learning that satisfies differential privacy, the pseudo-code for FLDP is provided in Algorithm 1. We consider a federated learning intrusion detection system with servers and clients. Each client $i$ has a set of data samples represented by $D_i$. The objective of the federated learning intrusion detection system is to train a machine learning model represented by the loss function $Loss(f(D_i;w))$ using the samples from the clients. Here, the $d$-dimensional $w \in \mathbb{R}^d$ represents the model parameters.

To protect data privacy, the process of the federated learning intrusion detection system is carried out through multiple global iterations by exchanging DP-noise-perturbed parameters between the clients and the server.

a) At the start of the global iteration $t$, the server sends the latest model parameters (represented by $w_t$) to some randomly selected clients (represented by $S_t$).

b) The selected client $i$ uses its private dataset $D_i$ and the latest model parameters $w_t$ to perform local iterations, obtaining the gradient $g_{i,t} + \text{Lap}\left(\frac{\Delta s}{\epsilon}\right)$, and generates the noise-distorted answer $a_{i,t}$, which is returned to the server.

c) The server aggregates the returned noise-perturbed parameters into $w_{t+1} \leftarrow w_t - \sum_{i \in S_t} \alpha_i a_{i,t}$, where $\alpha_i$ is the weight of client $i$, with $\alpha_i = \frac{|D_i|}{|D|}$. After the aggregation operation, the server begins a new round of global iterations with $w_{t+1}$.

**Algorithm 1** Differential Privacy-Compliant Federated Learning Algorithm

```
Input: N: The set of clients; T: The total number of global itera-
tions; ε: privacy budget; s: the maximum number of selected clients in
each iteration
Output: w_t: Aggregated model global parameters; Mask(w_t^d): Uploading the
position information of the global parameters
 1: Randomly initialize global parameters w
 2: for t ← 0 to T-1 do
 3:   Randomly choose at most s clients
 4:   for i ← 0 to s do
 5:     g_{i,t} ← argmin_w Loss(f(D_i; w_t)) \\ D_i represents the data of the i-
       th client
 6:     a_{i,t} ← g_{i,t} + Lap(Δs/ε) \\ The client randomly perturbs parameters and
       uploads them
 7:     a_t^s ← a_t^s + a_{i,t}
 8:   end for
 9:   w_{t+1} ← w_t − η_t (|D_i|/|D|) a_t^s \\ The server aggregates the returned noise per-
       turbation parameters
10: end for
```

## 4.2 Problem formulation

In the previous section, we described the overall framework of FLDP. However, this framework expends excessive privacy budget to protect unimportant parameters. Intuitively, if $g_{i,t}[j]$ is very close to 0, the gradient $a_{i,t}[j]$ can be considered negligible for model aggregation, as small DP noise can severely distort its value and degrade FL performance. Therefore, it only makes sense to return $a_{i,t}[j]$ for aggregation when it is far from 0. However, FLDP indiscriminately distorts all parameters without differentiating importance.

In VANETs with numerous devices, the triviality issue becomes more pronounced, wasting more privacy budget on protecting unimportant parameters. Our experiments on datasets from CAN-intrusion-datasets, CIC-IDS, and F2MD simulators under IID and non-IID distributions showed that fewer samples meant fewer parameters effectively altered per device. Under non-IID with 75 samples per device, over 97% of gradient absolute values were less than 0.01. Thus, in VANETs, excessive privacy budget is spent protecting trivial parameters, indicating it is particularly feasible to apply the Fisher Information Matrix to avoid budget waste.

## 4.3 Federated learning based on the Fisher Information Matrix

The client updates the sub-model using local data, which is similar to the model parameter update section in Algorithm 1 of traditional federated learning. The difference is that, at this point, the client only needs to update the parameters of the sub-model after pruning, rather than the entire model. At this time, the gradient update formula in Algorithm 1 can be rewritten in the following form: $w_{t+1}^u \leftarrow w_t - \eta_t \sum_{i \in S_t} \alpha_i a'_{i,t}$. Here, $a'_{i,t}$ represents the pruned model parameters.

From right image of Fig 1, after updating the parameters of the sub-model post-pruning, the client needs to send the new parameters back to the server. To satisfy privacy requirements during this process, we employ differential privacy protection methods to perturb the data to be uploaded. The perturbed data will carry randomness, making it difficult for the server to infer the original data of the client based on these parameters.

Finally, the server receives the parameters from the clients and aggregates them to update the model. Since the sub-models uploaded by each client are heterogeneous, this also requires

the clients to send additional position information for each parameter when uploading the model. The server will update all parameters in turn based on this information, thus obtaining a new model to complete the current iteration.

At the start of a new iteration, the server can send new sub-model parameters based on the position information from the previous round to each user without having to send the entire global model; or, to consider model accuracy stability, it can send the global model after a fixed number of iterations to enable each client to determine the new sub-model's location.

Moreover, from the perspective of communication costs, the data size required to upload position information is much smaller than that needed for uploading parameters. We only need to upload an additional mask matrix (position information matrix) equal to the total number of model parameters to represent the position information, and each dimension of this matrix is either 0 or 1 to indicate masking. This means that each dimension of the mask matrix only requires 1 bit of size; whereas typically, when using FLOAT32 data types, each dimension of the parameter matrix would occupy 32 bits. Thus, when the pruning rate $\rho > 1/32 \approx 3\%$, the FFIDS algorithm based on the Fisher Information Matrix can serve to save communication costs. In most cases, the pruning rate is far greater than 3%.

**Algorithm 2** Differential Privacy-Compliant Federated Learning with Fisher Information Matrix Algorithm (FFIDS Algorithm)

```
Input: Input: N: The set of clients; T: The total number of global
iterations; ϵ: privacy budget; s: the maximum number of selected cli-
ents in each iteration
Output: Output: wₜ: Aggregated model global parameters; Mask(wₜᵈ):
Uploading the position information of the global parameters
1: Randomly initialize global parameters w
2: for t ← 0 to T-1 do
3:    Randomly choose at most s clients
4:    for i ← 0 to s do
5:       gᵢ,ₜ ← argminₘLoss(f(Dᵢ; wₜ)) \\ Dᵢ represents the data of the i-
          th client
6:       Fᵢ,ₜ ← Σⁿⱼ₌₁(∂gᵢ,ₜ[j]/∂w[i]) \\ The importance indices of the parameters were
          computed based on the Fisher Information Matrix
7:       F(wᵢ,ₜ)←Esum₁≤ᵢ≤ᵈFᵢ] \\ The expected forms of the second-order
          moments for the component functions
8:       F(wᵢ,ₜᵒ) ← Sort F(wᵢ,ₜ) by decreasing order
9:       Mask(wᵢ,ₜᵈ) ← set pruning rate ρ and prune F(wᵢ,ₜᵒ) by ρ rate
10:      a′ᵢ,ₜ ← F(wᵢ,ₜ) · Mask(wᵢ,ₜᵈ) + Lap(Δs/ϵ) \\ The client randomly perturbs
          parameters and uploads them
11:      a′ˢᵢ,ₜ ← a′ˢᵢ,ₜ + a′ᵢ,ₜ
12:   end for
13:   aₜ ← Σˢᵢ,ₜ[a′ˢₜ/Mask(wᵢ,ₜᵈ)] \\ The server aggregates the returned noise
         perturbation parameters
14:   wₜ₊₁ ← wₜ − ηₜ Σˢᵢ |Dᵢ|/|D| aₜ
15: end for
```

In the parameter pruning of federated learning based on the Fisher Information Matrix as described in Algorithm 2, the client is tasked with the initial computation of the Fisher Information Matrix, denoted as $F_{i,t}$. For this purpose, the log-likelihood function is utilized as the loss function, expressed as follows: $L(Y|X; w) = \sum_{i=1}^{n} Loss(f(y_i|x_i; w))$. Here, $f(y_i|x_i; w)$ signifies the conditional probability of output $y_i$ given input $x_i$ under the parameterization of $w$. Subsequently, based on this loss function, the first-order derivatives for each parameter are calculated using Eq 1 to ascertain the importance indicator for each component of the parameter $w$, i.e., the quantified magnitude of each parameter. Ultimately, the required Fisher Information

Matrix is defined as the expected form of the second-order moment of the component functions: $F(w) = E[\Sigma_{1 \leq i \leq d} F_i]$, yielding the importance indicator $F(w_{i,t})$ for the model parameters of each client, where $E[\cdot]$ denotes the mathematical expectation. In practical applications, the arithmetic mean of the Fisher Information is computed by inputting a large volume of data samples, serving as an approximate estimate of the expectation, also known as the empirical Fisher Information. This information supplies a quantitative measure of the importance of each parameter.

Since the Fisher Information is defined with respect to model parameters, the pruning of neural networks is typically conducted at the neuron level. This necessitates that the client must holistically process all related parameters involving each neuron. Given that each neuron impacts the two weight matrices that connect it to the preceding and succeeding layers, two specific scenarios must be distinguished for analysis. Firstly, when addressing the weights of neurons in the input layer, all weights in the same row as the neuron must be taken into account, in accordance with the rules of matrix multiplication. Secondly, when dealing with the weights of neurons in the output layer, all weights in the same column as the neuron must be considered. After such computational processing, the client ranks each neuron's weight values, obtaining the sorted importance indicators $F(w_{i,t}^d)$ and the position information matrix for each parameter $Mask(w_{i,t}^d)$, and executes pruning on neurons with smaller weights based on these. Once the pruned model is uploaded to the server, the server recovers the pruned model parameters by dividing the uploaded model parameters $a_{i,t}'^s$ by the position information matrix $Mask(w_{i,t}^d)$, and then aggregates the model using $w_t - \eta_t \sum_i^s \frac{|D_i|}{|D|} a_t$ to obtain the aggregated parameters.

To optimize neural network performance, the implementation of neuron pruning is critical. The process involves the introduction of a pruning algorithm aimed at reducing network complexity through pruning. Post-pruning, the neurons and their corresponding weights that have been masked in either the rows or columns of the weight matrix will no longer participate in subsequent network operations, significantly reducing computational resource consumption. Moreover, at the conclusion of the neural network's inference or training process, the client only needs to upload the parameters of the pruned sub-model, thereby effectively alleviating the communication burden. A more exhaustive analysis and discussion regarding savings in computational resources and communication costs will be presented in the experimental section.

## 4.4 Privacy analysis

Given the inevitable reduction in the dimensionality of parameters (for $1 \leq i \leq d$) uploaded by the client in each round due to the Fisher Information Matrix mechanism, we set the pruning rate to $\rho \in (0, 1)$. Consequently, the volume of data that users are required to submit each round is reduced to $(1 - \rho)$ times the original amount. To elucidate how the Fisher Information Matrix mechanism can decrease the quantity of noise needed to be added under the same privacy budget, we first introduce the following lemma:

**Lemma:** In FL, if each dimension of the gradient is drawn from the same distribution, then applying the Fisher Information Matrix mechanism will reduce the real sensitivity from $\Delta s_1$ to $\Delta s_1(1 - \rho)$, where $\rho \in (0, 1)$ denotes the pruning rate.

**Proof:** According to the definition of Laplacian sensitivity $\Delta s = \max_{\forall w, D} \|s(D) - s(D')\|_1$, set $f_d \geq f_d'$ and define the upper bound $f_d = (y_1, y_2, \ldots, y_n)$ and the lower bound $f_d' = (y_1', y_2', \ldots, y_n')$, where $d$ represents the dimensionality of the gradient. Based on this, the definition of Laplacian sensitivity can be rewritten as: $\Delta s_1 = |y_1 - y_1'| + |y_2 - y_2'| + \ldots + |y_n - y_n'|$.

Considering that each dimension of the gradient follows the same distribution, we can assume $\Delta s_0 = y_1 - y_1' = y_2 - y_2' = ... = y_d - y_d'$.

Subsequently, substituting into Eq (16) yields $\Delta s_1 = s_0 d$. After applying the Fisher Information Matrix mechanism, the dimensionality is reduced to $(1 - \rho)d$, and the real sensitivity of the gradient becomes $\Delta s_1' = \Delta s_0 (1 - \rho)d = \Delta s_1 (1 - \rho)$, which verifies the aforementioned lemma.

In summary, the pruning technique reduces the consumption of the privacy budget. Put simply, when adding noise to the parameters to expend the privacy budget, the parameters not uploaded after pruning do not consume the privacy budget.

## 5 Experimentation

### 5.1 Datasets

During the evaluation phase of our study, we employed two public datasets targeted for external-network and in-vehicle network communication scenarios: CIC-IDS-2017 [17] and CAN-Intrusion-Datasets [16]. To evaluate the proposed algorithm in a broader range of environments, we utilized the publicly available F2MD [18] simulation framework to emulate real traffic scenarios and generate datasets. F2MD is an extension based on the VEINS framework, which itself is an open-source VANET simulation platform integrating the event-driven network simulator OMNeT++ [41] and the road traffic simulator SUMO. F2MD is capable of producing two types of datasets: the first is the VeReMi dataset, which records Basic Safety Message (BSM) information received by vehicles from nearby vehicles during transit, as well as whether the vehicle itself is acting as an attacker. The second type of dataset is BSMList, which contains the results of basic plausibility and consistency checks performed on each received message, provided by F2MD.

In processing these public datasets, to enhance the comprehensiveness and balance of the data, we performed feature extraction on each client and collected data. We then conducted random shuffling and balancing of the data to aim for a positive to negative sample data ratio of 1:1. The CAN-Intrusion-Datasets capture hacker activities within the vehicle network, including records of various attack tools used, and cover normal and attack patterns of CAN bus and external network traffic obtained through simulating car hacking attacks and Network Intrusion Detection Systems (NIDS) testbed setups. The CIC-IDS-2017 dataset, on the other hand, records communications between On-Board Units (OBUs) and Roadside Units (RSUs), encompassing a variety of attack types including Denial of Service (DoS), reconnaissance and injection attacks, gear spoofing, RPM spoofing, and fuzzing attacks. These datasets are crucial to our research because they contain up-to-date attack data that are highly suitable for detecting modern attack behaviors.

When using datasets generated by the F2MD simulator, we preprocessed the two datasets to better suit attack detection methods for training models. The specific steps include: first, calculating Dist and rDiff values and adding them to each message record in the VeReMi dataset; subsequently, merging the VeReMi dataset with the BSMList dataset. Specifically, we extracted the basic plausibility and consistency checks for each message from the BSMList dataset and matched them with the message records in the extended VeReMi dataset. Finally, we sorted the merged data based on information such as the vehicle ID of the message receiver, the pseudonym ID of the message sender, and the reception time.

### 5.2 Training model

From Fig 2, we have implemented a detection model wherein the private dataset $D_i$ of each client is input into a two-layer Bidirectional Gated Recurrent Unit (Bi-GRU) network,

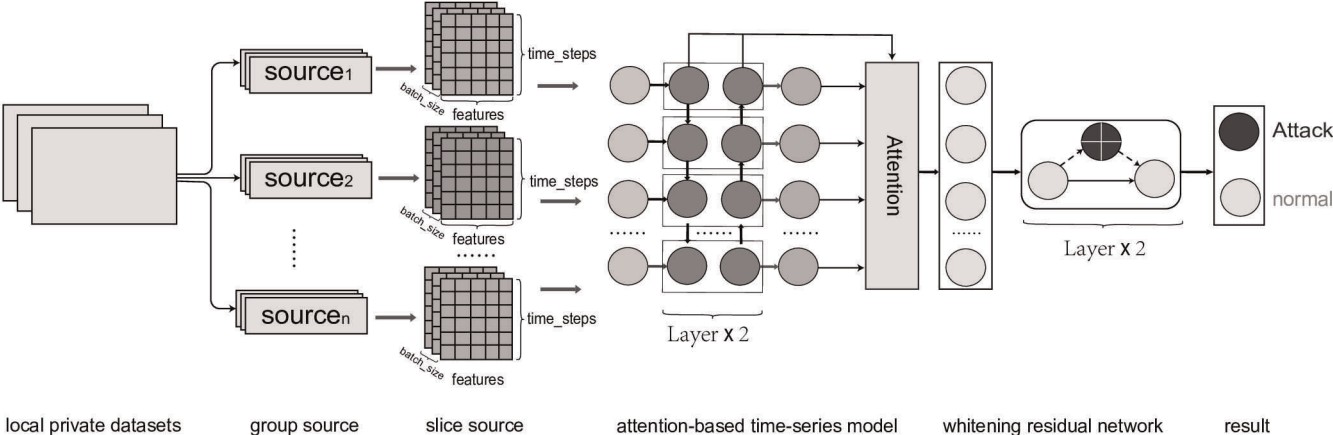

**Fig 2. Illustration of the process of the intrusion detection model.** Firstly, the local private data is grouped, and then these data are formatted into a time series format to meet the training requirements of the time series model. The formatted data is then inputted into a Multilayer Perceptron (MLP) to generate the final detection results.

incorporating the previous time step's hidden states $\overleftarrow{H_{t-1}}$ and $\overrightarrow{H_{t-1}}$. At each time step, the model outputs the hidden states $\overleftarrow{H_t}$ and $\overrightarrow{H_t}$, as well as the corresponding prediction result $O_t$.

We observed that the outputs $O_t$ generated at each time step contain rich information about the attack behaviors. To further enhance the detection accuracy and robustness of the model, we introduced an additive attention mechanism. This mechanism weights and accumulates the result outputs $O_t$, hidden states $\overleftarrow{H_t}$, and $\overrightarrow{H_t}$ at each time step, thereby integrating into the model training process to obtain the time step specific $S^i$. Subsequently, we applied the dropout function—a regularization technique used to mitigate the phenomenon of overfitting—to a three-layer fully connected neural network to obtain the final response representation $Re = Dropout(ReLu(S^i W_s + b)) + ReLu(S^i W_s + b)$, where $W_s$ is the weight matrix and $b$ is the bias term.

## 5.3 Experimental setup

From Fig 3A, vehicle-level FL model training was carried out on a Raspberry Pi 4B embedded device, which is equipped with a 64-bit dual-core Cortex-A72 CPU and 4GB of memory. From Fig 3B, the algorithm proposed in this study was executed within an Ubuntu/Linux 18.04 operating system environment, utilizing code written in Python 3.10, with the construction of deep neural network models facilitated by the API of the deep learning framework PyTorch 1.13. The development of the model and preliminary experiments were conducted on a computing platform equipped with an Nvidia RTX 3060Ti graphics card, featuring an 8-core Intel Core i7 10700F processor with a frequency of 2.9GHz and 32GB of RAM.

The training process for each model was carried out on the GPU using the Adam optimizer and spanned a total of 150 training epochs within a federated learning framework. Each client participating in the federated learning completed 10 training epochs locally. The initial learning rate of the model was set at 0.015, with a decay adjustment of multiplying the learning rate by 0.95 after each training epoch for each client. Specific parameters and hyperparameter settings of the model are detailed in Table 1.

In FL experiments, we selected 10 clients to act as clients in the FL process to ensure efficiency. To enrich the data available to clients, we first extracted the data features from all

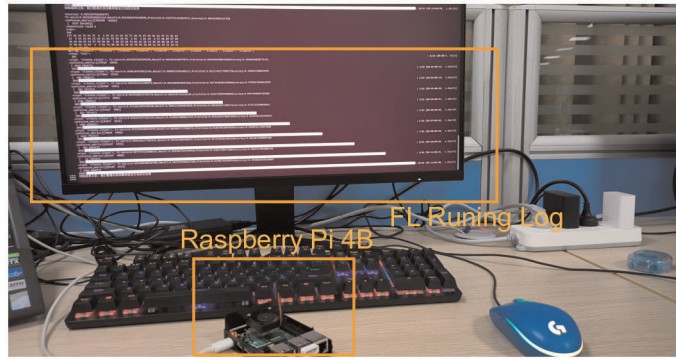
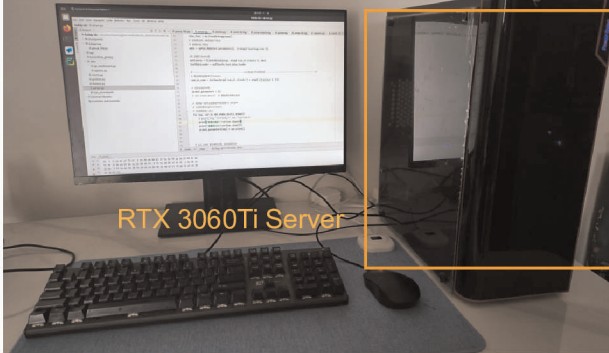

A. FL Vehicle-level Training                              B. FL Central Server Training

**Fig 3. Experimental setup.**

clients, then randomly shuffled and redistributed them to these 10 client vehicles. Considering the non-IID data distribution present in real-world scenarios, we allocated datasets of varying sizes to the vehicle clients and ensured that at least one vehicle contained only a single category label to simulate the non-IID attribute. It is important to note that the above data handling is solely for the purpose of simulation experiments; in actual scenarios, clients can only access their own local data for model training, without the need for aggregation and redistribution of data. Once the federated simulation commenced, these virtual local vehicle clients could only access their assigned data to train models, much like real clients conducting local training. The specific parameters for the federated learning are presented in Table 1.

### 5.4 Privacy budget

In the simulated real-world scenario, each vehicle client already independently possesses a portion of the data from the aforementioned four datasets, thus eliminating the need to receive distributed training data. Considering a privacy budget $\varepsilon \in [6, 8, 10, 14, 16]$, we evaluated a series of different values. Within this range, a lower privacy budget $\varepsilon$ provides stronger privacy protection for the vehicles.

### 5.5 Evaluation metrics

The article selected the F1 Score as the evaluation metric. The F1 Score is a very important performance measure because, unlike precision or recall which can bias towards one class, it provides a balanced performance indicator. The F1 Score is a commonly used metric in statistics

**Table 1. Federated learning for training model parameters and hyperparameters.**

| | |
|---|---|
| Round number | 150 |
| Client number | 100 |
| Number of clients selected for a round | 10 |
| Local clients batch size | 10 |
| Local clients epoch | 5 |
| Learning rate | 0.015 |
| Learning rate scheduler | 0.95 |
| optimization function | Adam |
| loss function | Cross Entropy Loss |

and machine learning to evaluate the accuracy of classification models, particularly in circumstances where the dataset is imbalanced. It is the harmonic mean of precision and recall.

## 6 Evaluation and discussion

### 6.1 Comparisons with the state of the art models

This paper conducts an in-depth comparative analysis of FFIDS against existing algorithms to showcase our research contributions. The compared algorithms will be detailed in the subsequent sections and are used as baselines for performance evaluation.

a) In the field of VANET intrusion detection, despite numerous outstanding research achievements, only FedMix [5] has considered the issues of privacy protection and communication costs. However, FedMix does not consider the perturbation of other parameters during the model upload process, and the FedMix algorithm simply selects a subset of client models for aggregation at the central server, leading to reduced accuracy. Other research primarily focuses on improving detection accuracy, yet overlooks the protection of model privacy and the challenges faced during actual communication processes. In comparison with existing work, this paper starts from the actual communication scenario of VANET, and by perturbing the parameters of the models uploaded by vehicle clients, it effectively reduces the risk of the models being reverse-engineered.

b) The research surrounding the DPFL-SR [12] is very stringent, wherein merely a solitary gradient update value is relayed in each global iteration. This work reveals that the convergence rate of the top-K algorithm is inversely proportional to K, described as O(1/K). In stark contrast, DPFL-SR manifests an exceedingly sluggish convergence when limited to a single gradient return. The FFIDS algorithm, on the other hand, leverages a positional matrix technique, empowering the server to swiftly consolidate the model parameters uploaded by clients. This strategic method accelerates convergence and enhances the accuracy of the model.

c) The FLDP algorithm proposed in LDP-FedSGD [10] integrates the Laplacian noise mechanism into the distributed machine learning model. As a special case of FFIDS, FLDP only contains a gradient noise perturbation module and does not perform selective perturbation when adding gradient noise, applying noise to all uploaded parameters. Compared to FFIDS, LDP-FedSGD consumes a privacy budget when returning minimal gradient updates, thus decreasing the accuracy of the federated learning model.

d) The NbAFL algorithm, proposed in literature [11], introduces the Gaussian noise mechanism into federated learning. In NbAFL, both parameters uploaded by clients to the parameter server (PS) and those distributed by the PS are affected by Gaussian noise. The Gaussian mechanism can only guarantee $(\epsilon, \delta)$-differential privacy, where $\delta$ represents the probability that NbAFL outputs do not satisfy $\epsilon$-differential privacy.

### 6.2 FFIDS performance evaluation

We initially set out to evaluate the impact of the FFIDS algorithm on model performance. In the relevant experiments, when training and evaluating FFIDS, we primarily compared the standard FFIDS algorithm, which lacks an integrated pruning technique, with the FFIDS algorithm that incorporates a random pruning technique. Additionally, we observed variations in model accuracy by adjusting the pruning rate. Analyzing from the perspective of pruning

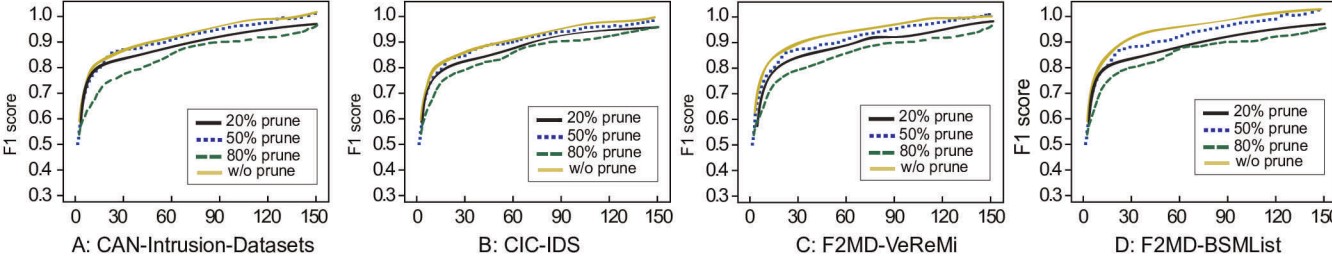

**Fig 4. Performance evaluation of the FFIDS algorithm.** Evaluation of different datasets with various pruning rates and 150 epochs iteration.

rates, as illustrated in Fig 4, the experimental results indicate that the FFIDS with a pruning rate set to 50% not only outperforms the 80% setting but, surprisingly, it also surpasses the 20% setting while closely matching the performance of the FFIDS algorithm without pruning.

Further, we explored the impact of the FFIDS algorithm on computational efficiency. Given that the FFIDS algorithm can effectively reduce computational costs only after integrating a pruning algorithm, and considering that Fisher information can only be computed after a complete model update, we must periodically refresh Fisher information after a certain number of epochs to adjust the neurons that need to be pruned.

## 6.3 Comparison of communication costs among different algorithms

In the experimental setup with a dropout rate of 50%, the update frequency of the Fisher information matrix was set to once per epoch (consistent with the experimental setup of the previous section), once every two epochs, and once every five epochs, for comprehensive testing. Under different pruning rates, the proportion of model pruning used in each training session remained consistent, ensuring that the pruning rate did not change mid-training. The communication cost considered in this paper refers specifically to the data transmitted by a single client during a round of federated learning training. Table 2 presents the communication cost incurred while training models on different datasets, where the FFIDS mechanism managed to train models with superior performance with less communication cost. For the

**Table 2. Communication cost of different models on different datasets and pruning rates.**

|  | LDP-FedSGD | | | | DPFL-SR | | | |
|---|---|---|---|---|---|---|---|---|
| pruning rate | CAN-IDS | CIC-IDS | VeReMi | BSMList | CAN-IDS | CIC-IDS | VeReMi | BSMList |
| 20% (kb) | 356+356 | 361+361 | 364+364 | 372+372 | 265+353 | 286+353 | 280+367 | 286+371 |
| 20% (kb) | 356+356 | 361+361 | 364+364 | 372+372 | 183+353 | 180+353 | 186+367 | 183+371 |
| 20% (kb) | 356+356 | 361+361 | 364+364 | 372+372 | 86+353 | 84+353 | 79+367 | 91+371 |
|  | NbAFL | | | | FedMix | | | |
| pruning rate | CAN-IDS | CIC-IDS | VeReMi | BSMList | CAN-IDS | CIC-IDS | VeReMi | BSMList |
| 20% (kb) | 363+363 | 372+372 | 368+368 | 369+369 | 345+345 | 358+358 | 369+369 | 374+374 |
| 50% (kb) | 363+363 | 372+372 | 368+368 | 369+369 | 345+345 | 358+358 | 369+369 | 374+374 |
| 80% (kb) | 363+363 | 372+372 | 368+368 | 369+369 | 345+345 | 358+358 | 369+369 | 374+374 |
|  | FFIDS | | | | | | | |
| pruning rate | CAN-IDS | CIC-IDS | VeReMi | BSMList | | | | |
| 20% (kb) | **295+295** | **301+301** | **286+286** | **295+295** | | | | |
| 50% (kb) | **189+189** | **183+183** | **189+189** | **190+190** | | | | |
| 80% (kb) | **82+82** | **84+84** | **83+83** | **101+101** | | | | |

communication cost measurement method presented in Table 2, we gauged the communication cost by directly measuring the size of the storage space occupied by the models uploaded from clients to the central server.

In exploring communication costs, our research reveals that within the FFIDS, the communication costs is minimized when the pruning rate is set to 80%. However, combining the data analysis results from Fig 4, the model performance under this setting is not ideal. Relatively speaking, a pruning rate of 50% not only yields the best model performance but also incurs lower communication costs than other model configurations.

Further comparing with DPFL-SR, in Table 2, we observe that in the case of a single model upload, the communication costs of DPFL-SR may be lower than that of FFIDS. However, when the total communication costs for uploading and updating models is accumulated, the cost of DPFL-SR is far higher than that of FFIDS. This finding aligns with our expectations, as the central server of DPFL-SR transmits the complete model data when distributing the aggregated model to clients.

Compared to other models such as NbAFL, LDP-FedSGD, and FedMix, during the federated training process, these models do not adopt a gradient clipping strategy. As a result, regardless of the pruning rate settings, the communication costs for these models remains at a fixed level.

## 6.4 Comparison of communication and time costs on FL vehicle-level training

To compare the efficacy of quantization-based gradient sparsification techniques, we deployed two distinct algorithms, cpSGD [42] and QuAsncFL [43], and utilized the gradient clipping-focused DPFL-SR algorithm as a reference for our control group. We meticulously trained each algorithm to ensure that they surpassed a threshold of 95% F1 score across various datasets. Furthermore, to closely replicate the performance characteristics of authentic vehicular embedded devices, we chose the Raspberry Pi 4B 4G as our testbed for vehicular-level experimentation.

On this platform, we conducted federated learning experiments for our intrusion detection model using the chosen gradient sparsification algorithms, which aids in evaluating the practical application effects of each algorithm in actual vehicular environments. Based on the data in Table 3, we can draw the following conclusions:

a) In vehicle-level federated learning training, although it is possible to significantly reduce communication overhead, the computational time is considerably extensive, far exceeding the time requirements (under 10ms) for intrusion detection systems (IDS) on embedded devices [44]. This confirms that by quantizing locally computed gradients and converting them into low-precision values rather than directly uploading the raw gradients, we can

**Table 3. Communication and time cost on FL vehicle-level training.**

| | FFIDS | | | | DPFL-SR | | | |
|---|---|---|---|---|---|---|---|---|
| Datasets | CAN-IDS | CIC-IDS | VeReMi | BSMList | CAN-IDS | CIC-IDS | VeReMi | BSMList |
| Communication Costs (kb) | **189+189** | **183+183** | **189+189** | **190+190** | 183+353 | 180+353 | 186+367 | 183+371 |
| Time Costs Per Iteration (ms) | **5.3-6.4** | **7.4-8.6** | **5.9-8.1** | **6.5-8.7** | 5.7-7.3 | 6.2-7.5 | 7.2-7.4 | 7.6-8.2 |
| | cpSGD | | | | QuAsyncFL | | | |
| Datasets | CAN-IDS | CIC-IDS | VeReMi | BSMList | CAN-IDS | CIC-IDS | VeReMi | BSMList |
| Communication Costs (kb) | 42.6+42.2 | 36.6+36.9 | 39.1+41.2 | 39.4+45.4 | 30.1+32.4 | 35.1+36.2 | 32.1+33.4 | 28.9+26.9 |
| Time Costs Per Iteration (ms) | 90.8-103.4 | 89.5-90.5 | 103.5-110.5 | 96.3+104.5 | 106.4-114.2 | 120.6-126.2 | 90.5-110.4 | 102.4-120.3 |

reduce the cost and bit count for each communication round. However, this approach sacrifices accuracy and hence increases the overall computational energy consumption, leading to a significant increase in time consumption in environments with limited computing resources on embedded devices.

b) We observed that the gradient clipping method used by DPFL-SR shows significantly lower computational time on embedded devices, all under 10ms. In contrast, although the FFIDS method proposed in this study can reduce communication loss during the upload phase, it does not perform model clipping during the download phase, resulting in no significant reduction in communication costs.

## 6.5 Comparison of model F1 score across different algorithms

In this experiment, we compared the baselines model F1 score of FL when combined with different DP algorithms. The baseline model FedMix, discussed in the previous section and lacking a DP mechanism, served as an upper reference for evaluation the impact of DP on model F1 score. Fig 5 presents the comparison of model F1 score on the test dataset for different algorithms during each global iteration. The default privacy budget was set to 10. To ensure fairness in comparison, we configured the same number of global iterations for LDP-FedSGD, FedMix, and NbAFL algorithms.

a) Without the interference of differential privacy noise, LDP-FedSGD achieved the highest model F1 score in all four experimental settings, with only a 2% to 8% difference in F1 score compared to the upper limit of the model trained by the FFIDS algorithm, which is much smaller than the gaps observed with other algorithms.

b) The model F1 score of FFIDS across four different dataset scenarios showed a significant improvement over DPFL-SR, especially in the F1 metric. This significant advantage is attributed to FFIDS being the first algorithm to implement gradient clipping in federated learning, which not only enhanced model performance but also demonstrated efficiency in consuming the privacy budget, proving to be more economical compared to other algorithms.

c) The performance of FFIDS also exceeded that of NbAFL. It is noteworthy that NbAFL offers a lower level of privacy protection than FFIDS because the Gaussian mechanism used by NbAFL only guarantees $(\epsilon, \delta)$-DP level privacy protection.

d) For the FedMix, despite the absence of differential privacy noise, FFIDS still maintained a lead in model performance. This is mainly because the central server only selected models with closer Euclidean distances for aggregation in the initial rounds of iterations, neglecting

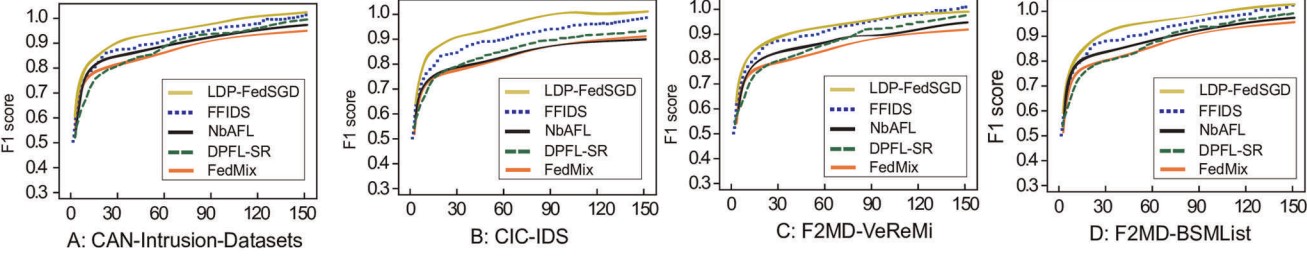

**Fig 5. Comparing the F1 score of different baseline models.** Comparing each global iteration with a fixed pruning rate of 50 with 150 epochs iteration.

potentially better-performing models that were further away, resulting in a significant decrease in overall performance.

e) LDP-FedSGD did not employ gradient pruning throughout the training process, as depicted in the revised Fig 5. The performance of LDP-FedSGD surpasses that of FFIDS, which is consistent with theoretical expectations. Inherently, when reducing communication over-head, a trade-off is made that inevitably sacrifices a certain degree of accuracy, leading to a decrease in the final model's performance.

## 6.6 Comparison of model F1 score under different privacy budgets

To comprehensively assess the model accuracy performance of various algorithms under different privacy budgets, a series of experiments were conducted. The values of the privacy budget $\epsilon$ were set at 6, 8, 10, 14, and 16, with corresponding adjustments made. During the experimentation, the implemented LDP-FedSGD algorithm did not incorporate differential privacy noise, therefore the F1 score of LDP-FedSGD remained unchanged when the privacy budget was altered. The experimental results are displayed in Fig 6, where the horizontal axis represents the privacy budget $\epsilon$, and the vertical axis indicates the accuracy of the final model after completing the FL process. Through the analysis of Fig 6, we can observe the following phenomena:

a) The magnitude of the privacy budget exerts a significant influence on the F1 score of the final model across all examined algorithms. As we constrict the privacy budget, a discernible decline in the model's F1 score is observed, indicating a trade-off between privacy preservation and model performance.

b) Within the spectrum of algorithms scrutinized for comparison, FFIDS invariably secures a position as the runner-up in terms of performance, reliably outstripping other DP focused algorithms in achieving superior model F1 scores under any given privacy budget configuration.

## 7 Conclusion

Although incorporating DP into FL clients can significantly improve privacy protection, the noise introduced by DP also impairs model accuracy. To mitigate the impact of DP noise on model accuracy and further improve the accuracy of FL models, we proposed an innovative federated learning algorithm with differential privacy called FFIDS, which integrates model parameter pruning techniques. FFIDS employs a pruning technique based on the Fisher information matrix to selectively preserve critical gradients in FLDP. This saves the corresponding

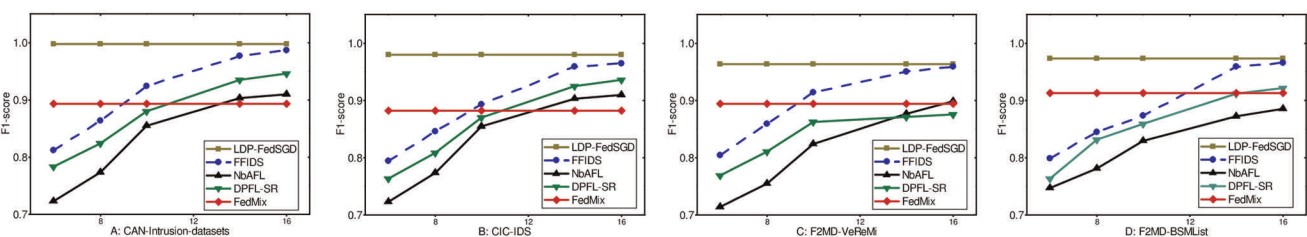

**Fig 6. Comparing the F1 score of different baseline models.** Comparing by varying privacy budgets from 6 to 16 with 150 epochs iteration.

privacy budget for numerous unimportant gradients. By comparing with other related studies, the superiority of the FFIDS algorithm is clearly demonstrated.

As a lightweight privacy-preserving training algorithm, FLDP is suitable for intrusion detection systems VANET, which has been validated through extensive experiments using both public datasets and simulator-generated datasets. Experimental results show that FFIDS has significant advantages over the current state-of-the-art benchmark algorithms. There is still much room for optimization and improvement of FFIDS. Future work will include combining other differential privacy mechanisms (such as Laplace distribution) with the Fisher information matrix and applying them to FLDP intrusion detection scenarios in VANET.

## Acknowledgments

We would like to sincerely thank the editors and anonymous reviewers for their helpful comments.

## Author Contributions

**Conceptualization:** Rui Chen.

**Data curation:** Rui Chen.

**Formal analysis:** Rui Chen.

**Investigation:** Xiaoyu Chen.

**Methodology:** Xiaoyu Chen, Jing Zhao.

**Software:** Rui Chen, Xiaoyu Chen.

**Supervision:** Jing Zhao.

**Validation:** Xiaoyu Chen, Jing Zhao.

**Visualization:** Rui Chen, Jing Zhao.

**Writing – original draft:** Rui Chen, Xiaoyu Chen.

**Writing – review & editing:** Rui Chen, Xiaoyu Chen.

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
