## [Decision Letter · Decision Letter 0]

4 Feb 2024

PONE-D-24-02004Sparsified Federated Learning with Differential Privacy for Intrusion Detection in VANETs based on Fisher Information MatrixPLOS ONE

Dear Dr. Chen,

Thank you for submitting your manuscript to PLOS ONE. After careful consideration, we feel that it has merit but does not fully meet PLOS ONE’s publication criteria as it currently stands. Therefore, we invite you to submit a revised version of the manuscript that addresses the points raised during the review process. Please submit your revised manuscript by Mar 20 2024 11:59PM. If you will need more time than this to complete your revisions, please reply to this message or contact the journal office at plosone@plos.org. Please include the following items when submitting your revised manuscript:A rebuttal letter that responds to each point raised by the academic editor and reviewer(s). You should upload this letter as a separate file labeled 'Response to Reviewers'.A marked-up copy of your manuscript that highlights changes made to the original version. You should upload this as a separate file labeled 'Revised Manuscript with Track Changes'.An unmarked version of your revised paper without tracked changes. You should upload this as a separate file labeled 'Manuscript'.If applicable, we recommend that you deposit your laboratory protocols in protocols.io to enhance the reproducibility of your results. Protocols.io assigns your protocol its own identifier (DOI) so that it can be cited independently in the future. For instructions see: https://journals.plos.org/plosone/s/submission-guidelines#loc-laboratory-protocols. Additionally, PLOS ONE offers an option for publishing peer-reviewed Lab Protocol articles, which describe protocols hosted on protocols.io. Read more information on sharing protocols at https://plos.org/protocols?utm_medium=editorial-email&utm_source=authorletters&utm_campaign=protocols.

We look forward to receiving your revised manuscript.

Kind regards,

Mohammed Balfaqih

Academic Editor

PLOS ONE

Journal Requirements:

Reviewers' comments:

Reviewer's Responses to Questions

**Comments to the Author**

1. Is the manuscript technically sound, and do the data support the conclusions?

Reviewer #1: Yes

Reviewer #2: Yes

2. Has the statistical analysis been performed appropriately and rigorously? 

Reviewer #1: Yes

Reviewer #2: Yes

3. Have the authors made all data underlying the findings in their manuscript fully available?

Reviewer #1: Yes

Reviewer #2: No

4. Is the manuscript presented in an intelligible fashion and written in standard English?

Reviewer #1: Yes

Reviewer #2: Yes

5. Review Comments to the Author

Reviewer #1: This paper introduces FFIDS, a novel approach for enhancing the security of VANETs through the integration of Federated Learning (FL) and Differential Privacy (DP). The primary focus is on addressing the challenge of preserving privacy in FL while minimizing the impact on model accuracy and communication costs. FFIDS employs a unique combination of model parameter pruning based on the Fisher Information Matrix and DP techniques. By identifying and pruning unimportant parameters, it reduces privacy budget consumption per iteration without sacrificing accuracy. DP noise is selectively added to submodels, prioritizing important parameters, thereby optimizing privacy preservation. The proposed approach is validated through extensive experiments on public and simulation datasets, demonstrating its effectiveness and superior performance in VANET intrusion detection models. However, some issues may need more attention.

1. Line 21 of the manuscript should be revised to replace "Inference attacks" with "member inference attacks." Additionally, there is a grammar mistake on line 14 that needs correction.

2. Line 189, should "g(w, D)" be changed to "s(w, D)"?

3. Line 211, should "Z[j]" be changed to "Z[i]"?

4. Line 218, should "wi" be changed to "w[i]"?

5. Is the loss function "P(y|x; w)" or "f(w)"?

6. In the experimental section, there are a few issues may need further considerations:

(1) Firstly, Table 2 should not exceed the length specified for the main text. Secondly, the data in Table 2 appears perplexing. Why are only NbAFL, DPFL-SR, and FFIDS included as methods? Additionally, there is no presentation of comparative results for communication overhead based on matrix update frequencies. Finally, I am particularly interested in how communication overhead is measured.

(2) In Figure 4, why does LDP-FedSGD outperform FFIDS? LDP-FedSGD does not involve parameter selection, and my understanding is that this approach should introduce more negative impacts. I am also puzzled by the fact that LDP-FedSGD in Figure 5 seems unaffected by the privacy budget.

(3) I suggest including a comparison of communication overhead with quantization-based methods, such as cpSGD.

Reviewer #2: This paper presents a privacy-preserving and communication-efficient Federated Learning algorithm for intrusion detection in VANETs. The primary concept revolves around utilizing the Fisher Information Matrix to evaluates the importance of model parameters, further pruning the model to retain important model parameters and adding DP noise only into these important parameters for diminishing the private budget consumption. Preliminary experiments are conducted to demonstrate the efficacy of the proposed approach on two public datasets and two F2MD simulated datasets. The suggested approach is acceptable, albeit with several critical concerns that require attention:

1)due to the involvement of gradient pruning in the proposed method, it is necessary to include relevant content in order to align the gradient pruning method with theoretical foundations in Section 3.

2)Please note that there is a error in Table 2 of the experimental results.

3)It is advisable to provide meaningful citation locations for references, for example, a method [21]. Please check for duplicate citations in the order of references.

4) Finally, please improve the quality of the paper by correcting grammatical errors and spelling mistakes, such as a spelling error in the left image of Figure 1.

Based on the comments above, I suggest making the necessary modifications before publication.

6. PLOS authors have the option to publish the peer review history of their article (what does this mean?). If published, this will include your full peer review and any attached files.

Reviewer #1: No

Reviewer #2: No

---

## [Author Response · Author response to Decision Letter 0]

6 Mar 2024

Dear Editor,

We are immensely grateful for the opportunity to revise our manuscript. It is with great pleasure that we submit the revised version entitled "Sparsified Federated Learning with Differential Privacy for Intrusion Detection in VANETs based on Fisher Information Matrix" for your consideration.

During the revision process, we have conducted all the suggested experiments, performed thorough analyses of related work, and made substantial clarifications where the content was previously unclear. We have addressed all the comments raised by the reviewers and yourself, which are elaborated on in the "Rebuttal Letter" attached to this correspondence. 

Aware of the high standards PLOS ONE upholds for article quality, we have also proactively refined areas not directly pointed out by the reviewers. These include enhancing the clarity of the introduction, and unifying the methodological explanations. The extent of these revisions can be seen in the "Revised Manuscript with Track Changes", also attached. 

All authors have read and consented to the resubmission of this manuscript. Should you have any questions or require further information, please feel free to contact me.

Thank you for your consideration of our paper and we are looking forward to hearing from you!

Yours sincerely,

Rui Chen

Dalian University of Technology

Email: chenrui_dut@163.com

---

## [Decision Letter · Decision Letter 1]

26 Mar 2024

Sparsified Federated Learning with Differential Privacy for Intrusion Detection in VANETs based on Fisher Information Matrix

PONE-D-24-02004R1

Dear Dr. Chen,

We’re pleased to inform you that your manuscript has been judged scientifically suitable for publication and will be formally accepted for publication once it meets all outstanding technical requirements.

Kind regards,

Mohammed Balfaqih

Academic Editor

PLOS ONE

Additional Editor Comments (optional):

Reviewers' comments:

Reviewer's Responses to Questions

**Comments to the Author**

1. If the authors have adequately addressed your comments raised in a previous round of review and you feel that this manuscript is now acceptable for publication, you may indicate that here to bypass the “Comments to the Author” section, enter your conflict of interest statement in the “Confidential to Editor” section, and submit your "Accept" recommendation.

Reviewer #1: All comments have been addressed

Reviewer #2: All comments have been addressed

2. Is the manuscript technically sound, and do the data support the conclusions?

Reviewer #1: Yes

Reviewer #2: Yes

3. Has the statistical analysis been performed appropriately and rigorously? 

Reviewer #1: Yes

Reviewer #2: Yes

4. Have the authors made all data underlying the findings in their manuscript fully available?

Reviewer #1: Yes

Reviewer #2: Yes

5. Is the manuscript presented in an intelligible fashion and written in standard English?

Reviewer #1: Yes

Reviewer #2: Yes

6. Review Comments to the Author

Reviewer #1: The authors have addressed all my previous concerns. Specifically, the corresponding symbols in the paper have been updated, and experiments have been supplemented promptly. The reviewer recommends an accept.

Reviewer #2: All comments have been addressed. The proposed approach is technically sound and the experimental results can suport the conclusion in this manuscript. I suggest this paper may be acceptable.

7. PLOS authors have the option to publish the peer review history of their article (what does this mean?). If published, this will include your full peer review and any attached files.

Reviewer #1: No

Reviewer #2: No

---

## [Editor Report · Acceptance letter]

4 Apr 2024

PONE-D-24-02004R1 

PLOS ONE

Dear Dr. Chen, 

I'm pleased to inform you that your manuscript has been deemed suitable for publication in PLOS ONE. Congratulations! Your manuscript is now being handed over to our production team.

Kind regards, 

on behalf of

Dr. Mohammed Balfaqih 

Academic Editor

PLOS ONE